# Surface-Enhanced Raman Spectroscopy for Molecule Characterization: HIM Investigation into Sources of SERS Activity of Silver-Coated Butterfly Scales

**DOI:** 10.3390/nano11071741

**Published:** 2021-07-01

**Authors:** Hiroyuki Takei, Kazuki Nagata, Natalie Frese, Armin Gölzhäuser, Takayuki Okamoto

**Affiliations:** 1Faculty of Life Sciences, Toyo University, Gunma 374-0193, Japan; 2Bio-Nano Electronics Research Centre, Toyo University, Saitama 350-8585, Japan; 3Graduate School of Life Sciences, Toyo University, Gunma 374-0193, Japan; kazu.1114.4289.sisimaru@gmail.com; 4Faculty of Physics, Physics of Supramolecular Systems, University of Bielefeld, 33615 Bielefeld, Germany; nfrese@uni-bielefeld.de (N.F.); ag@uni-bielefeld.de (A.G.); 5RIKEN, Saitama 351-0198, Japan; okamoto@riken.jp

**Keywords:** surface-enhanced raman spectroscopy (SERS), helium ion microscopy (HIM), silver-coated butterfly scale, categorization protocol of scales, structural modification, biomimetics

## Abstract

Surface-enhanced Raman spectroscopy (SERS) is a powerful technique for obtaining structural information of molecules in solution at low concentrations. While commercial SERS substrates are available, high costs prevent their wide-spread use in the medical field. One solution is to prepare requisite noble metal nanostructures exploiting natural nanostructures. As an example of biomimetic approaches, butterfly wing scales with their intricate nanostructures have been found to exhibit exquisite SERS activity when coated with silver. Selecting appropriate scales from particular butterfly species and depositing silver of certain thicknesses leads to significant SERS activity. For morphological observations we used scanning electron microscopes as well as a helium ion microscope, highly suitable for morphological characterization of poorly conducting samples. In this paper, we describe a protocol for carrying out SERS measurements based on butterfly wing scales and demonstrate its LOD with a common Raman reporter, rhodamine 6 G. We also emphasize what special care is necessary in such measurements. We also try to shed light on what makes scales work as SERS substrates by carefully modifying the original nanostructures. Such a study allows us to either use scales directly as a raw material for SERS substrate or provides an insight as to what nanostructures need to be recreated for synthetic SERS substrates.

## 1. Introduction

Surface-enhanced Raman spectroscopy (SERS) has been receiving increasing attention in recent years as techniques for fabricating requisite noble metal nanostructures mature and commercial SERS substrates become more readily available [1,2,3]. There are many applications, particularly in connection with biosensing, medical and environmental monitoring [4,5,6,7,8]. It is widely recognized that sharp points and edges on the size scale of tens of nm, ideally in a close proximity less than several nm of each other and a proper orientation with respect to the polarization direction of the incident laser, is desirable for effective SERS surfaces [9,10,11,12,13]. Many fabrication methods have been reported. Some have exploited bottom-up approaches such as attaching nano shells, nanorods, and nano stars onto a solid surface [14]. Some more elaborate approaches might exploit a metal film on nanosphere (MFON) structure whereby a noble metal is evaporated onto a single layer of surface-adsorbed nanospheres [15,16,17,18,19,20]. Chemical reactions such as reduction and galvanic reactions have also been used to form silver dendrites suitable for SERS applications [21,22,23,24]. Nano-porous metals are another fascinating approach [25]. Top-down approaches are represented by electron beam lithography which is capable of generating metallic nanostructures of almost any arbitrary shape [26,27]. In addition to these bottom-up and top-down approaches, one recent approach exploits natural microscopic structures such as wings of cicadas, dragonflies, and butterflies [28,29,30,31,32,33,34]. Wings of cicadas and dragonflies are characterized by sharp protrusions which, when decorated by nanostructured noble metal, function as a SERS platform. Butterfly wings in comparison are characterized by more intricate three-dimensional hierarchical microstructures ranging from nm to μm in dimension. Five major categories of structures have been reported [35], and a standard scale consists of a lower surface called lamina and an upper layer called lamellae. The former is flat and quite featureless, but the upper layer consists of parallel lamellae connected in the perpendicular direction by cross-ribs and supported by column-like struts. Butterfly scales have been made into SERS substrates either by metal decoration or replication. The former is referred to as bio-templating and the latter as biomimetics. Bio-templating may be carried out by thermal evaporation of a noble metal, reduction of noble metal ion into colloids by exploiting the reducing capability of hydroxyl and/or amino groups, or electrodeposition of copper by surface-adsorbed Au nanoparticles [31,36,37,38,39,40,41]. Biomimetics involves removal of chitin that constitutes the original butterfly scale after the replication process. These nanostructures have been shown to be effective SERS surfaces using model compounds, and their utility has also been demonstrated in the form of clinical assays for carcinoembryonic antigen (CEA) and malaria parasite [31,41]. Use of butterfly scales is, thus, a highly promising approach. For one, there are at least five groups of microstructures as mentioned above [35]. Even a single wing of one species includes scales with different microstructures. Of well over 150,000 known species of butterflies, many mostly contain scales that can be categorized into the above five groups, but there is a possibility that further investigations may lead to discoveries of additional new structures. Furthermore, each of these scales with a distinct microstructure can be further processed by various methods, leading to proliferations of SERS platforms to be examined. While butterfly scales are certain to offer useful insights for studies of the SERS effect, it is also possible to contemplate their direct exploitation. The chitin that makes up butterfly scales is stable, and scales can retain their physical integrity over many years as testified by excellent states of age-old butterfly collections. If they are to be used directly, one can obtain potentially thousands of scales from a single wing. A single scale more than suffices for a single measurement so that there is a possibility of greatly reducing the cost of SERS substrates. Typically, a single scale is 150 μm long and 100 μm wide so that a 1 cm by 1 cm area of a butterfly wing contains roughly 6000 scales.

While their utility as a potentially excellent SERS platform has been demonstrated, it is not completely clear why butterfly wing scales are effective. With scales coated with noble metal nanostructures, one can readily imagine that a corrugated surface or surface with sharp protrusions and nano-cone arrays characterizing the *Graphium* butterfly would allow the formation of densely adsorbed nanostructures over a larger surface area, which would serve as good SERS substrates [36]. On the other hand, the ridge-cross rib structure of *Eupleoa mulciber,* the striped blue crow, (henceforth referred to as *E. mulciber*) is relatively large, on the scale of hundreds of nm rather than tens of nm in dimension, thus not necessarily optimal for hot spot generation when coated with a metal as opposed to decoration with noble metal nanostructures. It is, thus, crucial to find out which part of the three-dimensional hierarchical microstructure makes significant contributions to the overall SERS activity. For this purpose, we first investigated standard, unmodified scales for their SERS activity. Then, we modified the standard scale to investigate its influence on SERS activity. More specifically, by using scales from *Sasakia charonda,* the great purple emperor, (henceforth referred to as *S. charonda*) and *Papilio thoas,* the thoas swallowtail, (henceforth referred to as *P. thoas)* characterized by a ridge-cross rib structure, (1) we varied the deposition thickness and (2) investigated polarization dependence. The former was an obvious way of affecting the morphology, and the latter was intended as a way to elucidate a role played by parallel ridge microstructures in the upper layer. Furthermore, (3) modified scales were studied. For this, we experimented with two approaches. In the first approach, we removed the upper layer with use of adhesive tape. By sandwiching a scale with two pieces of adhesive tape and forcefully separating them, it was possible to remove the upper layer. We evaporated silver either after removal whereby we obtained a lower surface uniformly coated with silver, or before removal whereby we obtained shadowed silver on the lower surface. In the second approach, we exposed scales to ozone. By varying the exposure time, we etched scales to varying extents. We also carried out morphology observation with a scanning electron microscope, SEM, and a helium-ion microscope, HIM. The former was used for overall characterization while the latter was used for more detailed observations. Use of the HIM allowed us to assess the metal-coated dielectric microstructure without having to apply an additional coating of an electrically conductive layer as is the case with SEM observation.

We found that with rhodamine 6G (R6G) as a model compound, there was a drastic increase in the signal intensity with the deposition thickness up to 20 nm. Further increasing the thickness to 100 nm led to only modest additional improvements. No polarization dependence was observed so that the lamellae and cross-ribs of the upper layer did not seem to play a significant role in the SERS effect. The limit of detection (LOD) was roughly on the order of 10^−6^ M for all types of scales from both *S. charonda* and *P. thoas*, but closer examinations showed that scales from *P. thoas* gave stronger signals than those from *S. charonda* at higher R6G concentrations, but the LOD of scales from *P. thoas* was higher than that of *S. charonda*. More interestingly, we found that removal of the upper layer from scales of *S. charonda* did not eliminate the SERS activity whether or not removal was carried out before or after metal deposition. The extent of the signal intensity reduction, however, did depend on the type of scales. When the area of the opening surrounded by ridges and cross-ribs, often referred to as window, was large as with chestnut scales from *S. charonda*, the reduction in the signal intensity was relatively small, but with white scales from *P. thoas*, the sequence of upper layer removal and metal evaporation made a difference.

As for effects of ozone treatment, we found that treatments up to 30 min enhanced the signal intensity, but further extending the treatment time led to a reduction in the signal intensity. With the exposure time of 120 min, the signal intensity declined to 20% of that of the untreated scale. To summarize, we conclude that there is more to butterfly scales than their three-dimensional hierarchical structure as far as SERS activity is concerned.

We also developed a sample preparation protocol which should be of use for investigations of butterfly scales in the future. Butterflies of different species are expected to possess scales with different microstructures, but even scales from a single wing of a particular species exhibit widely varying microstructures [37,42]. In order to distinguish one scale from another, it is tempting to use coloration as a differentiation index, but correlation between coloration and microstructure has not been proven. This could happen to scales with the same color within a single wing so that scales being explored must be characterized by coloration, shape, microstructure as well as precise location within the wing. As more and more species of butterfly come under investigation for SERS applications, random sampling from a wing is likely to lead to confusion so that it is imperative to establish at this stage a method to allow the tracing of individual scales to their original location [43]. While this requires more care in handling, the wing of even a single species contains a wealth of valuable information and it should be carefully combed for optimal microstructures rather than simply subjected to random sampling.

## 2. Materials and Methods

### 2.1. Materials

A private collection of *S. charonda* caught at various locations in Japan was provided to our group. Samples of *P. thoas* were purchased from the Nawa Insect Museum (http://www.nawakon.jp), Gifu, Japan. Scales transferred from wings were placed on a cover glass purchased from Matsunami Glass (Osaka, Japan. Cat. No. C022401), using a polyimide Kapton tape from Nitto Denko KK (Osaka, Japan. Cat. No. P223). Rhodamine 6G was purchased from Sigma-Aldrich (Tokyo, Japan) (Cat. No. R4127).

### 2.2. Sample Preparation

While most previous workers have used whole wings, we decided to transfer scales from a wing to a flat solid surface. A wing possesses vines which cause an uneven surface. Moreover, individual scales are not parallel to the base of the wing so that attaching a whole wing directly onto a solid surface would result in scales tilted at a certain angle to the underlying solid surface, requiring constant re-focusing of the excitation laser for SERS measurements. Thus, by transferring scales from the wing, we could reduce variations in the angle and height of individual scales under observation. We found that pressing a piece of natural rubber against scales on a wing resulted in transfer of scales onto the rubber. A similar approach with latex membranes has been reported [30]. Then, the rubber stamp was pressed against a glass slide covered by a piece of double-sided Kapton tape. This resulted in re-transfer of the scales onto the Kapton tape, ready for subsequent thermal evaporation. The resulting metal-coated scales were lying flat against the glass slide so that the need for refocusing was minimized upon lateral transfer.

We also paid special attention to keeping track of the original location of scales under investigation. It was tempting to think that scales could be simply categorized by coloration; in case of *S. charonda*, there were chestnut, cream, white, indigo, and orange scales and they seemed to be characterized by corresponding distinct microstructures. However, as we could not yet exclude the possibility that scales of the same coloration may possess different microstructures, we decided to keep track of the precise location from which particular scales were obtained. As a matter of fact, Janssen et al. reported on the relationship between scale structure and pigmentation in butterfly wings and concluded that there were no clear-cut correlations [43]. Thus, it is imperative to classify scales not only by pigment, i.e., coloration, but also by location within the wing and actual shape and structure. For this purpose, we adhered to the following protocol as shown in Figure 1. We excised circular specimens, 3 mm in diameter, from a wing by using a punch. They were randomly selected to include scales of different colors if possible. These circular specimens were placed on a glass slide covered by a piece of adhesive Kapton tape, (a). A rubber stamp, 3 mm in diameter, was pressed against each of the circular specimens to transfer scales from the wing onto the rubber surface. These scales were transferred onto another glass slide, also covered by Kapton tape. For the final thermal evaporation, we used the VFR-200M/ERH (ULVAC KIKO, Inc., Miyazaki, Japan) to evaporate silver pellets (Nilaco Co., Tokyo, Japan, 5N% Cat. No. 400025) at the vacuum pressure of 5 × 10^−3^ Pa or better. A typical evaporation rate was 0.2 nm s^−1^. The deposition thickness was monitored with a quartz crystal monitor CRTM-6000G (ULVAC KIKO, Inc., Miyazaki, Japan). Prior to thermal evaporation, a metal mesh with a hexagonal opening, with a 0.18 mm-long edge, (Cat. No. PA-23, Hasegawa Corp., Shizuoka, Japan), was placed as a shadow mask (b). With the mask in place, only areas of scales not covered by the mesh become metalized, easily distinguishable from non-metalized areas. By noting the location from which circular specimens were excised and the location of individual scales within the shadows cast by the mask, we could readily trace individual scales to their initial location within the wing. This allowed us to catalog all the scales we investigated, allowing us to correlate location, nanostructure, and SERS effectiveness. Unless described otherwise, the standard silver thickness was 100 nm.

### 2.3. Applications of Model Compounds

Unless described otherwise, scales were immersed in 1 mM R6G aqueous solution for 10 min. They were then gently rinsed with distilled water and dried.

### 2.4. Modifications to the Scales

#### 2.4.1. Physical Removal of the Upper Layer

A scale was placed on Kapton tape attached to a glass slide. Another piece of tape was placed on the scale, and a gentle but firm pressure was applied. The second tape was then lifted to remove the upper layer.

#### 2.4.2. Ozone Treatment

A butterfly scale was exposed to ozone generated from an ozone cleaner (UV 253E Filgen, Nagoya, Japan). By controlling the exposure time, we controlled the extent of structural modifications. The relative humidity also had an important effect on the etching process, higher humidity leading to more aggressive etching. For this, a shallow tray containing water was placed inside the ozone cleaner. A polystyrene well containing butterfly scales was placed in the tray. During the ozone treatment whose duration varied from 15 to 120 min, the flow of oxygen from an external cylinder was shut off.

### 2.5. SERS Measurement

For SERS evaluation, we used the Nicolet Almega XR, Thermo Fisher Scientific, with an excitation wavelength of 633 nm unless described otherwise. The aperture size was 100 μm with × 50 objective lens. The power setting of 670 µW was chosen to prevent photo damage. Whenever a new data set was to be taken for a particular type of scale prepared, two scales were probed at five randomly selected locations. The exposure time was 1 s, averaged over 16 measurements. For calculations of error bars showing the standard deviation, spectra obtained from five randomly selected spots of two separate scales, thus a total of ten spectra were used. For the investigation into polarization angle dependence, a single scale was selected and placed on a rotating stage. The relative angle between the polarization direction and the long axis of the scales was changed from 0 to 90 degrees at an increment of 30 degrees.

### 2.6. Scanning Electron Microscopy

For casual observation of microstructures, we utilized the Hitachi TM3000 (Hitachi High-Technologies, Corp., Tokyo, Japan). This has been designed as an electron microscope with the ease-of-use of an optical microscope, highly suited for observations of samples at magnifications slightly higher than normally possible with an optical microscope but with a good depth of focus. Its software allows automatic adjustment functions including auto-start, auto-focus and auto-brightness/contrast. The voltage was set at 15 kV. For more detailed observation, we used the Hitachi SU8030 (Hitachi High-Technologies, Corp., Tokyo, Japan) Ultra-high Resolution Scanning Electron Microscope, of the SU8000 Series Family. For most images, a secondary electron detector, angled at 45 degrees, was used. The accelerating voltage was set at 5 kV with working distance of 9600 µm. The angle of the incident electron beam is perpendicular to the specimen stage. For some images where so mentioned, SE-BSE signal mixing function of the Hitachi SU8030 was used to suppress charge up of poorly conducting samples. This unique feature allows a combination of secondary electron and backscattered electron signals at an arbitrary ratio, which was set at 50%.

### 2.7. Helium Ion Microscopy

HIM observations were carried out with a Zeiss Orion Plus^®^ (Oberkochen, Germany) with a helium ion beam of 34–36 kV acceleration voltage at a current of 0.5 pA. We used a 10 µm Aperture at Spot Control 5. Secondary electrons were collected by an Everhart-Thornley detector at 500 V grid voltage. The sample stage was tilted 45 degrees from the perpendicular unless described otherwise to allow observation of the nanostructure below the cross-ribs.

## 3. Results

### 3.1. Classification of Scales and their Morphology

*S. charonda* possesses five types of scale with different colors: chestnut, cream, chestnut, white, indigo, and orange. Their overall shapes can be grouped into three categories. Chestnut and cream scales have rounded apices whereas white and indigo scales have blunt apices. Orange scales possess a saw-toothed apex as shown at two different magnifications in Figure 2. The apex refers to the end of the scale away from the root where the scale is attached to the wing. We will refer to these scales as rounded, brunt and saw-toothed scales. At lower magnification, the overall shape of the scale can be readily recognized while the higher magnification allows observation of finer details, in particular cross-ribs spanning adjacent ridges.

Scales can be further categorized by different upper layers. While blunt scales (white and indigo) possess dense parallel ridges with gap distance of 1 µm; rounded scales (chestnut and cream) have the standard structure with ridges and cross-ribs forming windows, classified as Group I in ref. [35]. Ridges are joined by cross-ribs at an interval of 2 µm or so. The large window area was open. Orange scales are similar to the rounded scale in this respect. Chestnut and cream scales of *P. thoas* shown in Figure A1 in Appendix A are characterized by windows filled with porous patterns, referred to as Group III in ref [35].

### 3.2. Dependence of SERS Activity on Ag Deposition Thickness

We show in Figure 3a SERS spectra of 1 mM R6G obtained from chestnut scales of *S. charonda*, with 5, 20 and 100 nm silver deposition. These spectra as well as all the following spectra are shown without baseline correction to reveal the true spectra. The chestnut scale was selected as a representative scale of *S. charonda*, being most numerous for male and in particular female species. Each spectrum represents an average of spectra obtained from five randomly selected spots of two distinct scales, repeated three times for the total of 30 spectra, with the measurement condition described in the Experimental Section. The intensity of the 774 cm^−1^ band from the above spectra is represented as a bar graph in Figure 3b along with those obtained from cream, white, indigo, and orange scales of *S. charonda*, and chestnut and cream scales from *P. thoas*. Spectra from which band intensities were obtained are all shown in Figure A2. Both chestnut and cream scales of *P. thoas* give significantly stronger bands than any of the scales from *S. charonda*. Deposition thicknesses of 5, 20 and 100 nm were selected on the basis of a more detailed study of deposition thickness dependence shown in Figure A3. The signal intensity increased rapidly as the deposition thickness was increased up to 20 nm for chestnut scales from *S. charonda*, but depositing 40, 60, 80 and 100 nm of silver on the same scale led to only minor additional increases in intensity. Depositing 100 nm, however, did lead to the strongest signal so that we decided to use 5, 20 and 100 nm silver deposition in our standard protocol. A similar dependence was observed for the excitation wavelength of 532 nm as illustrated by Figure A3b.

While error bars in the bar graph of Figure 3b reflect variabilities of SERS spectra from these scales, we give more details in Figure A4. R6G SERS spectra in Figure A4a were obtained from five randomly selected spots within the central circled area of a cream scale shown in (b), and these spectra are superimposed. Intensities of five characteristic bands at 613 cm^−1^ (C-C-C ring), 774 cm^−1^ (C-H op bend), and those at 1361, 1505 and 1648 cm^−1^ (aromatic C-C) are shown as a bar graph in (c) [44]. We avoided taking data from peripheral regions where the ridge/cross-rib microstructures are significantly distorted with smaller windows. We further investigated the LOD by reducing the concentration of R6G. Scales were exposed to 1 mM, 100 µM, 10 µM, 1 µM and 100 nM solutions of R6G. Spectra obtained from chestnut scales of *S. charonda* are shown in Figure A5a. The bar graph in (b) summarizes intensities of the 774 cm^−1^ band from cream and white scales of *S. charonda* and chestnut and cream scales of *P. thoas*, based on the spectra shown in Figure A6. An LOD on the order of 10^−6^ M is similar to what has been reported earlier for SERS spectra of R6G with silver decorated biomimetic butterfly wing scales [28]. It is noteworthy that scales from *P. thoas* gave higher SERS intensities at higher R6G concentrations, but at lower concentrations, characteristic bands diminished in intensity more rapidly than from scales from *S. charonda* so that their LOD were higher. Moreover, at the lowest concentration of 100 nM some characteristic bands disappeared while some new bands appeared. At lower concentrations, characteristic bands diminished in intensity more rapidly than from scales from *S. charonda* so that their LOD were higher. Moreover, at the lowest concentration of 100 nM some characteristic bands disappeared while some new bands appeared.

### 3.3. HIM Characterization of Scale Morphology

Next, we looked into detailed structures of the upper layer in search of sources of SERS activity. Figure 4 shows HIM images of chestnut scales from *S. charonda* coated with silver of various thicknesses. They were taken from different angles around an axis perpendicular to the plane of the scale to facilitate visualization of the three-dimensional hierarchical microstructures. The ability of HIM to capture an image of poor electrical conductors such as non-metalized scale by simultaneous irradiation of neutralizing electrons was particularly advantageous as it eliminated the possibility of artifacts caused by coating of an electrical conducting material required for SEM observation. Under HIM observations, scales coated with 5 nm silver (not shown) did not look any different from non-metalized scales. When the deposition thickness was increased to 20 and 100 nm, however, silver deposited on the ridge could be recognized and they were distinct from each other. Even though the cross section of the ridge is close to 500 nm, thus not likely to contribute significantly to the SERS effect, chevron-like structures formed by lamellae may play a role in SERS activity.

### 3.4. Polarization Dependence of SERS Activity

To gain an insight into the role played by these ridge structures, we studied polarization dependence of SERS spectra. We rotated a scale with respect to the polarization direction of the excitation laser at an increment of 30 degrees. Figure A7 shows (a) images of a scale under observation and (b) the 774 cm^−1^ band intensity as a function of the angle of rotation. There was no clear angle dependence. This points to a possibility that parts of the microstructure other than the upper layer contribute significantly to the overall SERS activity.

### 3.5. Modified Scales

#### 3.5.1. Physical Removal of the Upper Layer

To investigate this possibility, we removed the upper layer by sandwiching a scale with two pieces of adhesive Kapton tape and forcefully separating them. This resulted in removal of the upper layer, revealing the lower surface without noticeable three-dimensional microstructure. For SERS measurements, silver was deposited either after or before removal of the upper layer. SEM images of such samples are shown in Figure 5a,b, corresponding to silver deposition after and before removal, respectively. What little could be observed was stump-like features from which struts had been pulled off. The silver-coated lower surface was immersed in 1 mM R6G solution and evaluated for its SERS activity. SERS spectra are shown below the SEM images. Interestingly SERS spectra are somewhat diminished in intensity in comparison to an unmodified scale but still quite prominent. It was difficult to remove the upper layer completely so that remnants still cover a large proportion of the scale. We made sure to irradiate only spots where removal was complete when obtaining these SERS spectra. Thus, stump-like features on the lower surface seen in the SEM image may make contributions to the overall SERS activity. One difference between Figure 5a,b is the presence of silver patches in the latter. Moreover, stump-like features are not likely to be coated with silver upon silver deposition in Figure 5b because the upper layer would shadow them. Figure A8 is a bar graph summarizing 774 cm^−1^ band intensities for chestnut and white scales from *S. charonda*, with silver deposition (a) before and (b) after removal. In (a), the difference in the signal intensity is striking, but in (b) the signal intensity is comparable for both chestnut and white scales, silver-coated stump-like features apparently largely contributing to the overall signal. We attribute this to the fact that the open window area of its upper structure, under 40%, is significantly smaller than hat of the chestnut scale, typically 60%, as can be assessed from the photos in Figure 2, so that the area of the lower surface covered by silver became correspondingly smaller.

#### 3.5.2. Ozone Treatment

Next, we show results of exposing scales to ozone before silver evaporation. This treatment resulted in relatively moderate modifications, at least for short exposure time. SEM images of scales treated with high-relative humidity ozone are shown in Figure 6. Images were obtained for the exposure times of 30, 60 and 120 min, shown at two different magnifications of 5000 (top) and 20,000 (bottom). After 30 min, the upper layer collapsed, with the supporting struts lying on their side. After 60 min, cross-ribs were significantly thinner; at 120 min, the upper layer was heavily damaged, leaving behind undulating ridge structures. Upon closer scrutiny, the chevron-like features seen in Figure 4 are no longer observed for exposure times longer than 30 min. SERS spectra of 1 mM R6G are shown for scales that were treated by ozone for different treatment time in Figure 7a, and the intensity of the 774 cm^−1^ band is shown as a bar graph in (b). There is an initial increase in the band intensity, more than doubling from the initial value, but after 30 min, the intensity begins to diminish significantly.

Figure A9 shows a SERS spectrum of 1 mM R6G of an orange scale from *S. charonda*, superimposed with a SERS spectrum obtained from a cream scale of the same butterfly. Bands other than those of R6G are circled. We did not observe them from other scales with different colors in this study. We speculate that it is associated with chemical species intrinsic to the orange scale, possibly from secretion. This can be taken as a warning to avoid certain scales for preparation of SERS substrates.

Our attempt to elucidate the source of the SERS activity is still at a preliminary stage, but one clearly needs to be careful in attributing the SERS activity solely to a complex three-dimensional microstructure. It has been attributed to sharp pointed structures [36], rib structures [37] and helices in gyroid structure [45]. When the surface has sharp pointed features, it can help increase the number of nanoparticles to be absorbed or form points upon metallization by evaporation or sputtering. Tan et al. considered sources of hot spots in faithful Cu replicas of scales of *E. mulciber* wing and concluded that rib-structures with a period of 20–30 nm on the sidewalls of main ridges, rather than the main ridges and struts themselves, played a key role [28]. These rib-structures are arranged in a periodic fashion in the direction perpendicular to the plane of the scale and hot spots were simulated in the narrow area between adjacent rib-structures. Lamella structures shown in Figure 4 and Figure A10 are also characterized by rib-structures similar to those of *E. mulciber* so that they may also play a similar role in generating hot spots. There are obviously other sources of hot spots and periodicity may be a key. Whereas cross-ribs of *E. mulciber* are arranged periodically in the direction along the main ridge, those of *P. paris* have specialized shape and are only quasi-periodic [42]. The former was found to have an LOD three orders of magnitude smaller than that of the latter, attributed to the higher structural periodicity. In the same paper, no anisotropy was reported with respect to SERS activity. This is consistent with our observation that there was no dependence on the polarization direction of the excitation laser. In addition, we also showed in Figure A5b that both chestnut and cream scales from *P. thoas* with quasi-periodicity had a higher LOD that those of *S. charonda* with more distinctive periodicity.

Tan et al. observed that, while the 3D bio-morphology was strongly responsible for SERS signals of R6G, a similar structure without the 3D feature gave rise to some signal, albeit one order of magnitude smaller [38]. They also explored various morphologies prepared by changing the deposition time and found that structures finer than main ridges and struts were likely to be responsible for the enhancement.

To the best of our knowledge, there has been no investigation into SERS activity of physically and chemically modified butterfly scales in order to experimentally confirm the source of SERS activity. While it is too early to conclude that the upper layer does not make the dominant contribution, one must also be careful not to be beguiled by its complex three-dimensional structure, a seemingly great source of hot spots. Sharp features in the scale less than 100 nm might be attractive for formation of hot spots upon metallization, but ozone treatment which has the effect of eliminating such sharp features has been shown to enhance the SERS activity. Thirty-minute exposure more than doubled the signal intensity. After 120 min exposure to ozone, the upper layer became obliterated, and the lower layer became exposed. Here, the SERS activity is significantly diminished so that ozone may affect the surface characteristics of the lower surface that contributes to SERS activity.

The enhancement performance of this biomimetic SERS substrate is quite respectable, but we also must mention some potential weaknesses. In the literature, many researchers report exceptionally high reproducibility of signal intensities from butterfly scale-based SERS substrates, but often there is no description of experimental conditions under which such data were obtained. In our experience, one must definitely focus the excitation laser in the central area of an individual scale as the microstructure around the peripheral region are distorted and give variations in the signal intensity. Another has to do with relatively high background due to the imbedded pigment. Thirdly, if quantitative results are desired, one must be careful in picking scales from particular locations within a wing. Random sampling of scales, even of a similar color, can results in wide variations in the signal intensity.

On the positive side, a large variety of butterflies, beside *S. charonda* and *P. thoas*, is capable of giving similar enhancement effects when coated with silver. Even older scales from butterflies that have died a natural death can be equally useful. It is safe to say that as long as the excitation laser can be focused on the central region of a scale with the aid of a microscope, one can expect semi-quantitative signals of reasonable intensity, comparable to some commercial products.

We are currently carrying out comprehensive comparison tests of butterfly-scale based SERS substrates against SERS substrates prepared with other methods, including commercial products. As discussed by Bell et al. [46], standardizing ways to evaluate and compare different SERS substrates still remains a challenge. Even calculating enhancement factors is not a trivial matter, dependent on various parameters. While it is still an ongoing project, the advantage of exploring butterfly scales probably lies with its relatively high sensitivity, suited for qualitative applications, and the fact that they can offer a whole variety of relatively well-ordered submicron structures that cannot be easily imagined.

## 4. Conclusions

This paper is not so much about the peculiarities of *S. charonda* and *P. thoas* as potential SERS platforms, but rather a general procedure for evaluating butterfly scales; these two species serve merely as examples. We have developed a protocol that can specify precise locations of individual scales in relation to the whole wing while a single scale can be selected for simultaneous SERS, SEM and HIM characterizations. When this protocol was followed, we found scales of a particular color possess had highly similar microscopic structures. When coated with silver in the range of 20 to 100 nm by thermal evaporation and subsequently exposed to R6G solutions, they all showed sensitivity on the order of 1 µM or less. The thickness needed to be at least 20 nm, but further increases up to 100 nm resulted in only modest improvements in the signal intensity. For performance comparison among different scales, it is necessary to pay attention to the concentration of the model compound, R6G here, because greater peak intensities at higher concentrations do not automatically translate into better LOD as demonstrated by comparison among scales from *P. thoas* and *S. charonda*. The former exhibited greater bands for 1 mM R6G but lower peaks for 1 µM in comparison to the latter.

The contribution of the upper layer was explored by investigation into the polarization dependence, i.e., whether linear silver microstructures on the ridges and cross-ribs may show anisotropy. There was, however, little dependence on the rotational angle, suggesting that other microstructures contribute to the SERS activity. This was amply demonstrated when the upper layer was completely removed. Silver deposited before as well as after removal of the upper structure exhibited ample SERS activity, particularly for the latter case. We speculate that stump-like features, formed by silver deposited on the root area of uprooted struts are responsible for the SERS activity. The ozone treatment had the effect of removing sharp features so that it was contrary to our initial expectation that treatments of any duration would diminish the activity.

We have not been able to pick the most crucial component of the butterfly scale so that simulations of the near field with a method such as FDTD calculation are somewhat premature at this stage. It is likely that different parts, both within the upper layer and the lower surface, make contributions. The variability in the signal intensity we observed is somewhat greater than what has been previously reported by other groups. Nonetheless, the sensitivity is significant, and they are sufficient for semi-quantitative measurements. More importantly, they offer opportunities to evaluate a whole variety of nano/microstructures, and careful combing through them promises to generate fascinating microstructures suited for SERS application. It is hoped that the protocol discussed is this paper will facilitate evaluations of butterfly scale-based SERS substrates.

## 5. Patents

We have a pending patent on a process to align scales with the correct orientation for the optimized SERS effect.

## Figures and Tables

**Figure 1 nanomaterials-11-01741-f001:**
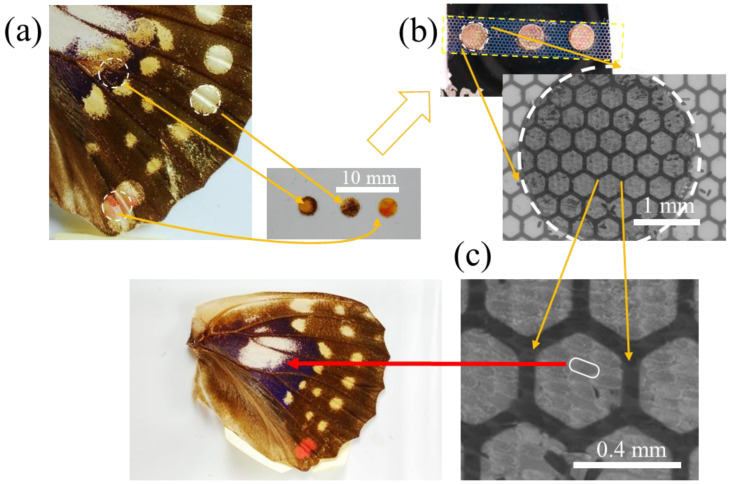
The protocol for cataloging individual scales. (**a**) Circular specimens excised from a wing of *S. charonda*. (**b**) Excised specimen transferred onto a glass slide and silver deposited through a metal mesh. (**c**) Individual scales traceable to initial locations within the wing. Scale bars: (**a**) 10 mm, (**b**) 1 mm and (**c**) 0.4 mm.

**Figure 2 nanomaterials-11-01741-f002:**
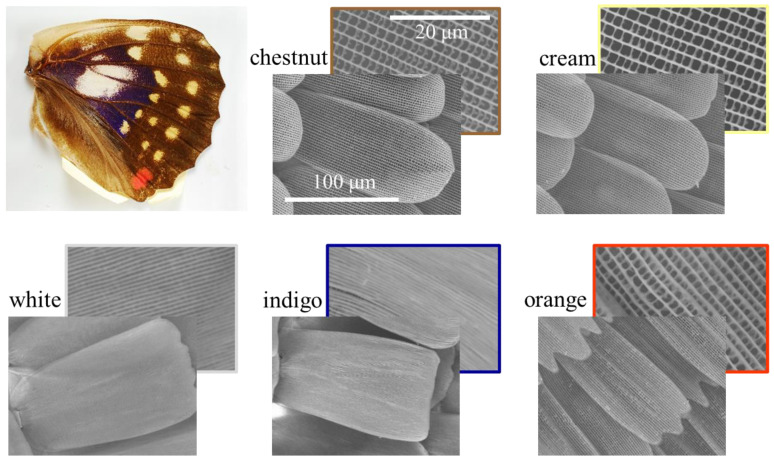
SEM images (Hitachi TM3000) of five types of scale from *S. charonda* as classified by coloration, corresponding to chestnut, cream, white, indigo, and orange scales. Images at lower magnification (scale bar: 100 µm) show the overall shape, and those at higher magnification (scale bar: 20 µm) reveal microstructures consisting of ridges and cross-ribs.

**Figure 3 nanomaterials-11-01741-f003:**
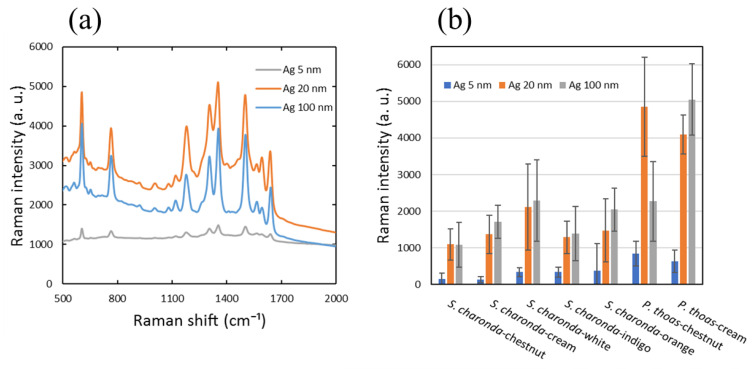
SERS activities of various scales of *S. charonda* and *P. thoas.* (**a**) SERS spectra of 1 mM R6G from chestnut scales of *S. charonda*, with 5, 20 and 100 nm silver deposition. (**b**) Bar graph summarizing intensities of the 774 cm^−1^ band from chestnut, cream, white, indigo, and orange scales of *S. charonda,* and chestnut and cream scales of *P. thoas*. Spectra from which the intensities were calculated are shown in Figure A2.

**Figure 4 nanomaterials-11-01741-f004:**
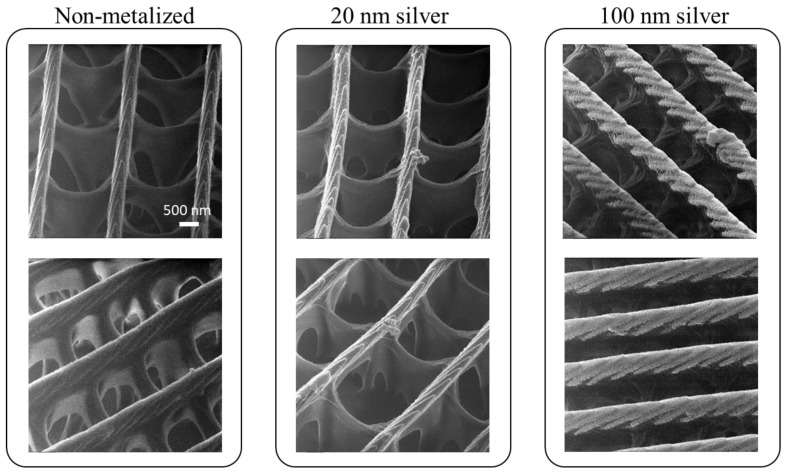
HIM images of chestnut scales from *S. charonda* with various silver deposition thicknesses. These were taken from different orientations and angles to facilitate the visualization of the three-dimensional hierarchical structures. The tilt angle is 45 degrees for all, expect for the top left image where it is 30 degrees. Scale bar: 500 nm.

**Figure 5 nanomaterials-11-01741-f005:**
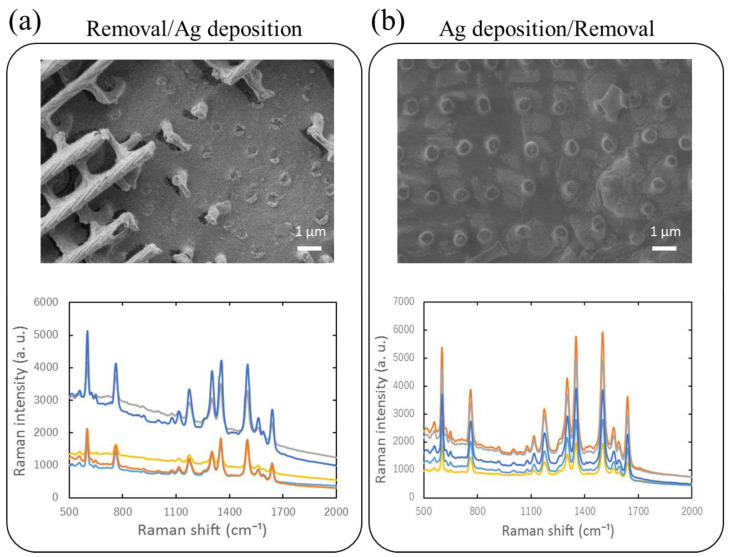
SEM images (Hitachi SU8030) of chestnut scales from *S. charonda* with the upper layer removed, (**a**) before and (**b**) after silver deposition. The SE-BSE mode was employed. SERS spectra of 1 mM R6G are shown below.

**Figure 6 nanomaterials-11-01741-f006:**
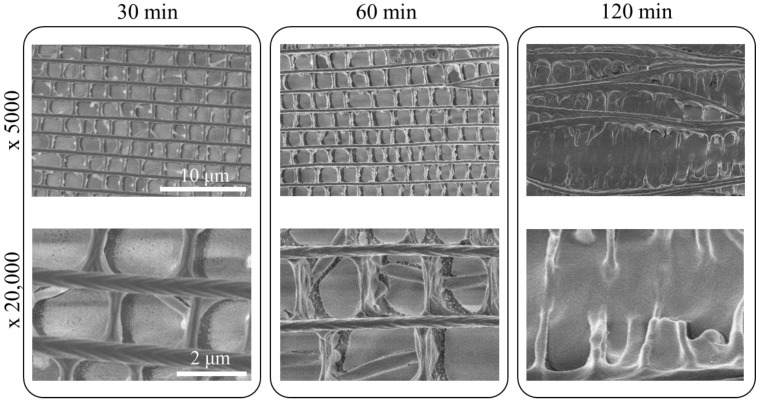
SEM images (Hitachi SU8030) of chestnut scales treated with ozone for 30-, 60- and 120-min. Scale bars: 10 µm (upper row) and 2 µm (lower row).

**Figure 7 nanomaterials-11-01741-f007:**
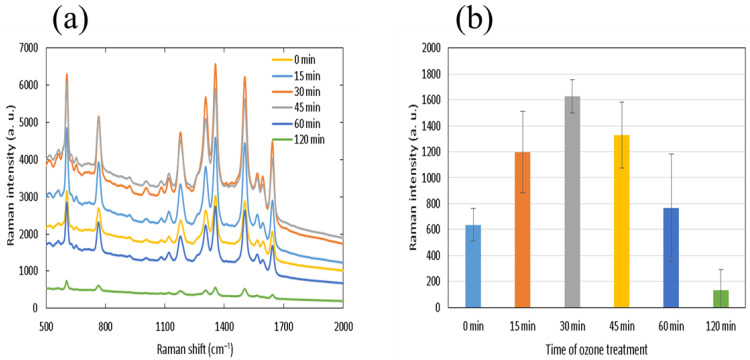
Effect of the ozone treatment on the SERS spectrum. (**a**) SERS spectra of 1 mM R6G after various exposure times, 15, 30, 45, 60 and 120 min. (**b**) the dependence of the 774 cm^−1^ band intensity on the treatment time. The error bars correspond to the standard deviations of signals obtained from five randomly chosen spots from two scales.

## Data Availability

Data is contained within the article.

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
