# Peer review of "Surface-Enhanced Raman Spectroscopy for Molecule Characterization: HIM Investigation into Sources of SERS Activity of Silver-Coated Butterfly Scales"

_nanomaterials, 2021, doi:10.3390/nano11071741_

Round 1

Reviewer 1 Report

General Comments

The study of Takei and coauthors deals with the description of a new protocol for SERS using scale wings of butterflies. The experimental design included different techniques such as SEM, HIM and SERS. The description of new methodologies is a crucial question in all scientific fields. Although the research presents interesting aspects, the experimental design and manuscript fails in some crucial aspects, decreasing the quality of this research. These aspects must be addressed by the authors. Overall, authors need: (i) to detail methods, especially electron microscopy conditions; (ii) to add statistical analyses of quantitative data; (iii) to add similar SEM images of all samples; (iv) to use adequate SEM analytical techniques (EDX, EELS) to enhance and to reinforce the results of silver localization; (v) to explain main advantages and disadvantages of the technique; and (vi) to stress in novelty of technical aspects, especially those of related to applied studies; and, finally (vii) to revise English and to correct mistakes. In my opinion, there is a high interest due to the a apparently high novelty and high accuracy of the data. I consider with high interest for readers and I would think it adequate for publication in a top journal as Nanomaterials after adequate revision.

Specific Comments:

  • Add entire scientific name (and common name) of each species used in the study.
  • Add detailed information of materials and methods, especially in electron microscopy techniques (sample preparation, microscope conditions, detector) and check differences of quantitative data with adequate statistical analyses at the end of materials and methods section.
  • Add similar information of electron microscopy images in all samples at two magnifications. Check if all images are made at same conditions of magnification, working distance to the sample and angle of inclination between sample and detector.
  • Add a semiquantitative analysis and mapping images with EDX or EELS to detect amount and localization of silver at nanoscale in samples.
  • Correct mistakes such as in line 18 (leads leads).
  • Add a brief description of advantages ad disadvantages of the method in the discussion.

Author Response

Thank you very much for taking your precious time for useful advice.

  • We have included the entire scientific name as well as the common name of the butterfly species we have studied. Graphium butterfly and mulciber, from references, are left as they are.
  • For electron microscopy we have added more information we have on hand. Unfortunately we do not have the information on the angle of inclination between sample and detector. In this paper, we did not resort to statistical analysis of SERS spectra for the following reason. Our categorization of various scales according to the color and submicron structures might not be sufficient yet to differentiate different types. And as we pointed out, the spectrum has a definite dependence to the precise location within a single scale. In the near future when we are convinced that we have done our best for categorization, we will carry out statistical analyses of each type of scales so categorized.
  • All the images in Fig. 2 were obtained with the TM3000. It is a fully-automated microscope, we do not have any information other than what has been added. By the way, we made use of the TM3000, TM8030 and HIM in accordance with convenience and information required. The TM3000 located within the same building as our laboratory is most readily accessible, good enough for characterization of scales with a full view. For more detail as in Fig. 6, we resorted to the TM8030 located on another campus 1 hour drive away. For the most critical characterization, samples prepared in Itakura were shipped to Bielefeld. There was obviously a consideration for the available hours for each machine.
  • We are certainly interested in obtaining images with EDS or EELS in the near future as we do have access to such instruments on an affiliated campus where the TM8030 is also located. It is hoped that such analysis would help us create a proper model for finite difference time domain simulations of the associated near-field.
  • I have added some more explanation. Briefly, butterfly scales offer a diverse body of nanostructures for forming SERS substrates. Even though the signal uniformity is not its forte, its sensitivity is rather respectable, exceeding most commercial products, which we carried out in connection with another study. I will ask the editor whether it is advisable to include such data. The diversity of nanostructures is likely to lead us to better nanostructures with extensive comparison.

Reviewer 2 Report

In this manuscript the authors performed an extensive investigation on the use of butterfly scales as SERS substrate. Silver is used to coat the scales (3 different thicknesses are reported) and to enable field localization on the nanostructures surface.

The manuscript is well written and, to me, rather interesting.

Several details are discussed and the figures are clear and explicative.

Before to recommend the publication, anyway,  it should be interesting to improve some aspects

  1. being an organic substrate (scales), can the authors show some examples of the background in the measurements? I expect some peaks from the substrate (that can be cancelled with a thick enough layer of metal)
  2. the reproducibility is a major issue here. the community working on SERS is now looking for stable and reproducible platforms, here it is clearly not the case. This is mentioned, but some strategy to improve the reproducibility could be mentioned

In the introduction the authors mentioned several strategies to prepare SERS substrate. Among the others they could mention nanoporous metals. With respect to others, these latter can be prepared without lithographic steps and show good stability and reproducibilty. (see for example DOI:10.1021/acsnano.0c10945; DOI:10.1002/adma.201706755)

Author Response

Thank you very much for taking your precious time for useful advice.

  • The background is an important issue. We have obtained spectra from samples coated with silver but not yet exposed any Raman probe. Under our conditions of SERS measurements described in the experimental section, most scales do not show artifacts, but the fluorescence background as can be seen also with Fig. 3 (a) is not negligible. Our extensive evaluation of commercial products from six vendors, outside the scope of this paper, has revealed artifacts, but in general relatively high fluorescence background is a weakness of butterfly scale SERS substrates. Obviously, some give more and others give less fluorescence background, and this would be an important criteria for picking right scales from right butterflies. The suggestion of using a thick metal layer is very interesting. We have tried something similar with coating of a dielectric layer to keep the intrinsic surface away from the deposited silver. This seems to have the effect of reducing the background.
  • Some earlier reports on the butterfly scale-based SERS substrates have claimed extraordinary signal intensity uniformity, but we have not been able to show that in this paper. We can imagine a number of ways to increase the uniformity. (1) Certain butterfly species might possess scales of higher uniformity so that we will continue evaluating more species, (2) Fragmentation of scales to sizes smaller than 1 µm and homogenization of these fragments into a single continuous film, (3) Further processing of scales along the line of ozone processing.
  • We have included one of the papers on nanoporous metals you suggested. It is indeed an interesting system, and we are working on a paper on a similar system.

Reviewer 3 Report

See attached file.

Author Response

Thank you very much for taking your precious time for useful advice.

  • Providing EF would definitely improve the quality of this paper as you suggest. We have not done so in this manuscript for the following the reason. For one, we are still in the process of screening for appropriate butterfly species. After evaluation of some more species, we will pick the best one for more careful characterization including calculations of EF values. Such evaluation is expected to involve additional Raman probes beyond R6G at various concentrations. For now, I have added Ref. 45 that emphasizes standardization of the evaluation protocol that includes calculations of EF.
  • For now, it is true that only Fig. A3 includes spectra obtained with a laser line different from 633 nm. We have begun evaluation of butterfly scale-based SERS substrates with other Raman probes such as BPE, crystal violet, and some pesticides such as ferbam and TBZ at various concentration. We would like to make them part of our next paper as it is expected to take some time to finish this evaluation. We are also using other excitation wavelengths such as 532 and 785 nm.

Round 2

Reviewer 1 Report

Authors included common and scientific name of some species  Papilio thoas, Thoas swallowtail), others are not correctly named (Sasakia charonda Hewitson) and others remained with genus and common names (E. mulciber).

About information of electron microscopy, authors improved this part of materials and methods. About missing information, I am sure authors could contact with microscopy facility staff to obtain required information. I aprreciate the response of statistical analyses but I consider not completely true. Figures 3B and 7B are derived from quantitative data that must be analyzed with adequate statistical procedures. All quantitative data expressed as a mean and a bar of standard error or deviation must be compared with adequate statistical tests. Without statistical analyses, it is not possible to define or discuse differences, decreasing the qualituy and accuracy of the study.  These aspects are mandatory for a correct study.

Missing EDX or EELS analyses decrease the quality and interest of manuscript for readers but it is not a crucial aspect for the acceptance of the manuscript.

Author Response

I have contacted the electron microscope facility for further information on various detectors on the Hitachi TM8030. It is hoped that the additional information in blue is sufficient. 

We have also revised names of the butterflies we used and made a reference to, again shown in blue.

As for statistical analyses, I am not sure whether you were simply pointing out that we did not define the error bar and the number of sampling, or maybe something more. At least for the definition and the number of sampling, we have added the missing information.

Thank you very much for your advice. 

Reviewer 3 Report

The revised version of the manuscript can be accepted for the publication.

Author Response

Thank you again, for taking your time.

Round 3

Reviewer 1 Report

Authors need to revise carefully the new text of detectors. In the present form, the text have not sense and it is not the way to obtain the images from this manuscript.

in the present form, names of butterfly species is correct

Statistical analyses are just comparisons between 2 or more groups. As previously commented, authors need to compare their quantitative data (detailed in previous revision) with adequate tests (i.e. ANOVA, t-student, Kruskal-Wallis, Mann-Withney or other adequate tests).

I revised 3 times this manuscript and I accepted non crucial aspects for enhance quality and interest of the manuscript for reader. However, adequate electron microscopy and statistical methods are mandatory for a correct study. Without adequate response of these two aspect, I consider this article not acceptable for publication in Nanomaterials.

Author Response

Thank you for your thorough reviewing. We have managed to get hold of an English-language technical manual for the Hitachi SU8030. Using this material, we have modified a part of the experimental section shown shown in red. Apparently, it is a fairly unique feature of this particular SEM, not widely used; I have copied a small part of it and attached it. It should be of help to the reviewer to see if our explanation makes sense or not. 

As for additional analysis, we do not recall analysis such as ANOVA, t-student, Kruskal-Wallis, Mann-Withney  etc. employed in SERS-related papers. When we went through all the papers cited in our paper, we fail to see it used; at the very least most papers treat their data by simple calculations of standard deviations. Nor have I ever encountered such analysis in a number of manuscripts I have reviewed on behalf of MDIP. Such analyses may possibly be useful, but in view of the fact that no single approach is widely used, it may not warrant extra effort required.
